# Comparison of In Vitro Bacterial Susceptibility to Common and Novel Equine Wound Care Dressings

**DOI:** 10.3390/ani14050776

**Published:** 2024-03-01

**Authors:** Merrill Simpson, Dean A. Hendrickson, Doreene R. Hyatt, Sangeeta Rao

**Affiliations:** 1Department of Clinical Sciences, Colorado State University, Fort Collins, CO 80523, USA; 2Department of Microbiology, Immunology, and Pathology, Colorado State University, Fort Collins, CO 80523, USA

**Keywords:** wound dressing, bacterial susceptibility, minimum inhibitory concentration (MIC), minimum bactericidal concentration (MBC), Manuka honey, hypertonic saline, polyhexamethylene biguanide (PHMB), silver sulfadiazine

## Abstract

**Simple Summary:**

Wound care is a challenging part of equine veterinary practice and there are a variety of wound dressings and strategies to manage wounds. Our goal was to ascertain the relative efficacy of honey when compared to common wound dressings as it pertains to reducing bacterial growth. This experiment was performed in a laboratory setting where different species of bacteria were grown in a simulated wound environment. These bacteria were then exposed to medical-grade Manuka honey, local honey, and commercial, food-grade honey as well as other commonly used wound dressings to see how effective each wound dressing was on reducing bacterial growth. Our results showed that polyhexamethylene biguanide (PHMB, a commonly used wound dressing) was best at reducing bacterial growth. Local honey out-performed Manuka and store-bought honeys. Interestingly, the most susceptible type of bacteria was harvested from an actual equine wound when compared to the lab-grown bacteria. We suspect that the complex wound environment plays a large role in determining the efficacy of wound dressings. Our results provide data to practitioners so they can decide how to best manage an infected wound based on the type of bacteria growing.

**Abstract:**

Antimicrobial resistance is becoming a problem of concern in the veterinary field, necessitating the use of effective topical treatments to aid the healing of wounds. Honey has been used for thousands of years for its medicinal properties, but in recent years medical-grade Manuka honey has been used to treat infected wounds. The goal of this study was to determine the relative susceptibility of four common equine wound pathogens to ten different types of antimicrobial agents based on the minimum inhibitory concentration (MIC) and minimum bactericidal concentration (MBC). The pathogens studied include ATCC lab-acclimated *Pseudomonas aeruginosa*, *Escherichia coli*, and methicillin-resistant *Staphylococcus aureus* and one from an equine sample submitted to the Colorado State Veterinary Diagnostic Laboratory (*Streptococcus equi* ssp. *zooepidemicus* (*Streptococcus zooepidemicus*)). An additional goal of the study was to describe the comparison of bactericidal activity of medical-grade Manuka honey, local honey, and commercial, food-grade honey to other commonly used wound dressings (20% hypertonic saline, silver sulfadiazine cream, PHMB gauze, and PHMB foam). The objective is to provide veterinary practitioners with comparative data on the use of a variety of antimicrobial dressings for inhibiting the growth of common wound bacteria. MIC and MBC for Manuka, store, and local honeys were comparable to those of sterile gauze, sugar, and hypertonic saline. Across bacterial species, local honey proved to have more bactericidal activity when compared to Manuka honey and commercial, food-grade honey. The MIC and MBC for PHMB gauze and foam was consistently at a higher dilution compared to the other antimicrobials. The majority of antimicrobials exhibited stronger inhibitory and bactericidal activity against a Streptococcus zooepidemicus isolate obtained from a wound compared to other bacteria that were ATCC lab-acclimated. Additional research for in vivo applications needs to be done to see whether differences exist in effective wound management.

## 1. Introduction

There is a high incidence of traumatic wound infections in equine patients and an increasing prevalence of antimicrobial-resistant bacterial species in wounds [1,2]. A 2015 report published by the United States Department of Agriculture Animal and Plant Health Inspection Service (USDA APHIS) indicated that 16.3% to 27.8% of equid deaths (20 years of age and under) nationwide were the result of skin wounds or trauma [3]. A 2018 prospective study on antimicrobial resistance in horses in the UK showed that the prevalence of multi-drug resistance to *E. coli* was 31.7% and for *Staphylococcus* spp. it was 25.3% [4]. The nature of horses and their habitat puts them at risk of wounds that are likely to become contaminated with soil bacteria [5]. Horses often suffer wounds on their legs, and these wounds are more predisposed to delayed healing due to tension and infection [5,6]. The overuse of antibiotics is a significant contributing factor to increasing bacterial resistance [5]. Researchers are investigating alternative therapies to help reduce the need for systemic antibiotics in both human and animal wound infections [5]. Topical treatments reduce the development of resistance to systemic antibiotics, are cost-effective, and decrease morbidity [7,8]. Topical antimicrobials affect multiple targets and reduce the use of prophylactic antibiotics, and avoid side effects commonly seen with systemic antibiotics [7,8]. A better understanding of topical antimicrobial agents in animals is important to provide the most successful wound healing.

Biguanides are an important class of antimicrobial agent, used for the preservation of many aqueous compounds such as disinfectants and antiseptics [9]. One commonly used biguanide is chlorhexidine digluconate (CHG); the widespread use of CHG has led to the development of new biguanides such as PHMB [9]. There are many different types of wound dressings impregnated with PHMB (polyhexamethylene biguanide), and these dressings help by decreasing bacterial load and preventing bacterial penetration, therefore reducing colonisation of the wound bed [7]. PHMB is a broad-spectrum, yet safe, antimicrobial used as a post-surgical dressing [10]. In a systematic review of the effectiveness of PHMB, topical PHMB has been shown to reduce bacterial burden, promote healing, and alleviate wound-related pain in people [11]. PHMB is effective against a broad spectrum of Gram-positive and Gram-negative bacterial species as well as fungi and yeast [12]. PHMB has been shown to be efficacious against methicillin-resistant *Staphylococcus aureus*, *Escherichia coli*, *Pseudomonas aeruginosa*, and 12 other bacteria [12,13]. As an antimicrobial, PHMB is bactericidal, reducing bacterial colonization and invasion while maintaining a balance of normal skin flora [13]. PHMB antimicrobial dressing is commercially available as a gauze or foam dressing [12].

Hypertonic saline (20%) dressings are used to debride infected or exudative wounds [14]. These dressings work by removing fluid from bacteria and necrotic cells using osmotic gradients, allowing bacteria and debris to be lifted off a wound with dressing change [14]. In a study comparing hypertonic saline (3%) to normal saline (0.9%) for wound irritation, the researchers found that hypertonic saline was better in the initial stages of healing but was shown to have negative outcomes once granulation tissue forms, likely due to desiccation of the healthy granulation tissue [15]. Hypertonic saline has been proven to help debride heavily exudative wounds when combined with negative pressure wound therapy in people, helping to control infection and reduce the need for surgical debridement [16].

Silver sulfadiazine (SSC) preparations are also commonly used for topical wound treatment, although the exact mechanism of action is currently unknown [17]. The topical application of 1% SSC to incisional wounds in rats is associated with faster wound healing versus that in control wounds [18]. Treatment with 1% silver sulfadiazine cream (SSC) resulted in improved neovascularization and epithelialization compared with the results for 0.25% sodium hypochlorite, bacitracin, and 10% povidone–iodine solutions in mice with injuries [19]. Silver sulfadiazine has proven to reduce microbial counts in horses with metatarsal and metacarpal injuries; however, it also resulted in exuberant granulation tissue that required surgical resection [20]. A study comparing SSC and honey in vivo found that the use of honey resulted in the reduction of early inflammatory signs, better control of infection, and quicker wound healing when compared to SSC [21].

Advancements in wound care have been made including the use of medicinal Manuka honey as an antimicrobial and anti-biofilm agent in chronic wound management [5,13]. The use of medical-grade Manuka honey for the treatment of resistant bacterial infections has shown promising success [22]. Medical-grade Manuka honey has proven in vitro bactericidal activity against *Bacillus subtilis*, *Escherichia coli*, *Pseudomonas aeruginosa*, and methicillin-resistant *Staphylococcus aureus*. Manuka honey also has demonstrated potent in vitro activity against antibiotic-resistant bacteria and has been shown to be successful in the treatment of wound infections not responding to antibiotic therapy [22,23]. The efficacy of honey for wounds can be attributed to its bactericidal compounds, a moist environment that promotes healing, and high viscosity that maintains an effective barrier [22]. The antimicrobial properties of honey in general include hypertonicity, low pH, and hydrogen peroxide production, but these properties are negligible with dilution [22]. Phenolics are compounds found in honey that act as antioxidants and contribute to the anti-inflammatory and healing properties of honey [24]. Honey aids autolytic debridement and promotes the growth of healthy granulation tissue [21]. This selective destruction of bacterial cells is what separates honey from many of the other topical antimicrobials, which have some inherent cytotoxic effects [25]. By inadvertently destroying the healthy cells that are trying to re-populate in the wound bed, wound healing is slowed with some topical products [25].

It was discovered that Manuka honey maintains its antimicrobial effect even with dilution and has “non-peroxide activity” due to a compound called methyl glyoxal (MGO) [24]. MGO results from the spontaneous dehydration of its precursor dihydroxyacetone (DHA), a naturally occurring phytochemical found in the nectar of flowers of *Leptospermum scoparium*, *Leptospermum polygalifolium*, and some related *Leptospermum* species native to New Zealand and Australia [24]. There is no evidence of damage to host cells; however, the mechanism behind this selective toxicity to bacterial cells is not understood [24]. Manuka honey has been tested in vitro against many types of skin and wound pathogens, and notably shows efficacy against multi-drug-resistant (MDR) bacterial phenotypes and bacteria in biofilms [24]. Medical-grade Manuka honey has proven efficacy and is tested to be free of contaminants and then pasteurized to ensure the product is safe for medical use [25]. Manuka-type honey has been shown to have anti-biofilm activity as well [26]. Within chronic wounds, bacteria within a biofilm are protected from host defenses and harbor antimicrobial resistance [27]. New Zealand Manuka-type honeys in wound dressings prevent and eradicate *Staphylococcus aureus* biofilms. Methylglyoxal (MGO), dihydroxyacteone (DHA), and hydrogen peroxide are the antimicrobial and anti-biofilm components in Manuka honey [26]. Medical-grade honey has proven immunomodulatory properties that facilitate wound repair [26].

Horses have a tendency toward chronic non-healing wounds or exuberant granulation tissue [6]. Topical antimicrobials such as Manuka honey have proven in vivo to create more organized granulation tissue, accelerating healing [28]. In vivo studies using Manuka honey in wounds in equine distal limbs show that there is improved angiogenesis and a more mature granulation tissue bed characterized by low inflammation and improved epithelialization when compared to untreated wounds [28]. In one study there was no difference in aerobic and anaerobic bacterial counts between wounds treated with Manuka honey and untreated wounds, despite the known antibacterial activity of Manuka honey [28]. The authors of this study cited the possible reason for this being that the Manuka honey was applied just once a day, which may not be frequent enough for maintenance of antibacterial activity in the wound environment [28].

The goal of this study was to provide veterinary practitioners with comparative data on the use of various dressings, including PHMB, hypertonic saline, medical-grade honey, commercial food-grade honey, and local raw honey, to inhibit the growth of common bacterial species in wounds. We ascertained bacterial susceptibility in vitro to gather information for possible future in vivo utilization. Our hypothesis is that medical-grade honey would have a comparable bactericidal effect to PHMB and would have a better bactericidal effect than commercial food-grade honey and local raw honey. We also hypothesized that hypertonic saline would have similar bactericidal effects to PHMB.

## 2. Materials and Methods

Medical-grade Manuka honey was used from the brand Kruuse AD (Absorbent dressing) (Jørgen Kruuse A/S, Langeskov, Denmark) which is a sterile honey-impregnated absorbent dressing with 100% *Leptospermum scoparium* honey from New Zealand [29]. Store-bought honey was Kroger brand clover honey (U.S. Grade A, Product of USA, Canada, and Argentina) [30]. The local honey was harvested in Pierce, Colorado, in 2015. Apiculturists would classify this honey as wildflower honey, as the hives are not deliberately placed near a single source of nectar, which is common in commercial apiculture [31]. The majority of the nectar contributing to this honey is likely from alfalfa plants [31].

PHMB-impregnated, 0.5% Kendall AMD (Cardinal Health, Dublin, OH, USA) foam and PHMB-impregnated, 0.2% Kerlix AMD (Cardinal Health, Dublin, OH, USA) gauze was used [12].

The bacterial isolates for testing were from the American Type Culture Collection (ATCC) (*Pseudomonas aeruginosa* 27853, *Escherichia coli* 25922, and methicillin-resistant *Staphylococcus aureus* 25923) and from an equine sample submitted to the Colorado State Veterinary Diagnostic Laboratory in April 2016 (*Streptococcus equi* ssp. *zooepidemicus (Streptococcus zooepidemicus))*. Isolates were grown on trypticase soy agar plates with 5% sheep red blood cells overnight at 37 °C, and isolated colonies were suspended in phosphate-buffered saline to a 0.5 McFarland standard using the BBL™ Prompt™ (Becton, Dickinson and Company, Franklin Lakes, NJ, USA) inoculation system (BD). This resulted in a concentration of bacteria at approximately 1.5 × 10^8^ colony forming units per ml. Test materials included sterile gauze with 20% hypertonic saline, 1% silver sulfadiazine cream, PHMB gauze, PHMB foam, local raw honey, commercial, food-grade honey, and medical-grade Manuka honey. A solution of 20% hypertonic sugar was used as control for the honey to assess if hypertonicity was responsible for any antimicrobial activity that might be observed.

In terms of antimicrobial susceptibility testing, two dilutions of each bacterial species were used to determine the MIC and MBC with challenge doses of bacteria. The 0.5 McFarland solution was used undiluted and was diluted with Mueller Hinton Broth to reach approximately 10^5^ CFU/mL. To determine the minimum inhibitory concentration (MIC), a constant volume of 50 microliters of the bacterial dilution was added to doubling dilutions of each antimicrobial treatment at a constant volume of 100 microliters (diluted in 0.9% saline) in a 96-well plate. The plates were covered and incubated overnight at 37 °C. After incubation, the plates were read visually for the minimum concentration of the drug that was required to inhibit visual growth of the organism. To determine the minimum bactericidal concentration (MBC), all wells of the incubated plate that had no visible bacterial growth were subcultured to trypticase soy agar plates with 5% sheep red blood cells and incubated overnight at 37 °C. The minimum concentration of antimicrobial that allowed no bacterial growth on subculture was recorded.

Due to the opacity of SSC, it was not possible to interpret any reduction in bacterial growth in the 96-well plate; therefore, no MICs could be determined. To observe MBC based on an 80% reduction in growth, a lawn of each bacterial species was made on individual sheep blood agar plates, while 10 microliters of each dilution of SSC was pipetted onto the agar plates and incubated overnight at 37 °C. The zone of inhibition, a circular area around the antimicrobial where the bacteria does not grow, was used to measure the susceptibility of the bacteria to the antimicrobial [32]. The amount of growth in the zone of inhibition was recorded for each bacterial species.

## 3. Results

Total sample numbers are based on two dilutions of each bacterium, four organisms tested, 10 treatment antimicrobials and 10 replicates for MIC and MBC. Each bacterial species was tested with PHMB gauze and foam, hypertonic saline, Manuka honey, local honey, and commercial, food-grade honey at 10^5^ and 10^8^ CFU/mL because these represent typical bacterial counts in contaminated wounds. Testing for SSC was only against bacteria at 10^8^ CFU/mL.

Table 1 and Table 2 show the minimum inhibitory concentration of each wound dressing represented in the columns with each bacterial species at either 10^5^ or 10^8^ CFU/mL in the rows. The dilution of an antimicrobial that resulted in no visible growth of bacteria in a 96-well plate is the minimum inhibitory concentration for that antimicrobial with a given bacterial species. 1:1 = original concentration of the antimicrobial, 1:2 = 50% of the original concentration, 1:4 = 25% of the original concentration. The percentages next to dilutions are the percentage of samples where bacterial growth was inhibited for a given dilution out of 10 replicates. NE = Not examined. MIC for SSC was examined for each bacterial species at 10^8^ CFU/mL only. The MIC for SSC was not able to be visualized due to the opacity of SSC.

Table 3 and Table 4 show the minimum bactericidal concentration of each wound dressing represented in the columns with each bacterial species at either 10^5^ or 10^8^ CFU/mL in the rows. The dilution of an antimicrobial that resulted in no visible bacterial growth when subcultured to trypticase soy agar plates with 5% sheep red blood cells is the minimum bactericidal concentration for that antimicrobial with a given bacterial species. 1:1 = original concentration of the antimicrobial, 1:2 = 50% of the original concentration, 1:4 = 25% of the original concentration. The percentages next to dilutions are the percentage of samples where bacterial growth was inhibited for a given dilution out of 10 replicates. NE = Not examined. MBC for SSC was examined for each bacterial species at 10^8^ CFU/mL only. No MBC means that growth was visible on all agar plates when subcultured and incubated overnight at 37 °C.

The MICs and MBCs for Manuka, store-bought, and local honeys were comparable to those of sterile gauze, sugar, and hypertonic saline, at 1:1 to 1:2 for all bacterial species tested. Across bacterial species, local honey proved to have more bactericidal activity when compared to Manuka and commercial, food-grade honey. Local honey had an MBC at a dilution of 1:1 for *Pseudomonas aeruginosa* and *Staphylococcus aureus* (10^5^ CFU/mL), while commercial, food-grade honey and Manuka honey had no MBC, which means there was visible growth on all subcultures. Manuka honey and commercial, food-grade honey had an MBC for *Streptococcus zooepidemicus*, ranging from 1:1 (90–100% of samples depending on the isolate) to 1:2 (10% of samples).

The MIC and MBC for PHMB gauze and foam were variable among different bacterial species and proved to have a stronger bactericidal and inhibitory effect on *Escherichia coli* and *Streptococcus zooepidemicus* than on *Staphylococcus aureus* and *Pseudomonas aeruginosa*. The MIC and MBC for PHMB gauze and foam were consistently at a higher dilution compared to the other antimicrobials. The majority of antimicrobials exhibited stronger inhibitory and bactericidal activity against a *Streptococcus zooepidemicus* isolate that was obtained from a wound compared to the other organisms that were ATCC lab-acclimated. PHMB gauze had bactericidal effects against *Streptococcus zooepidemicus* (10^5^ CFU/mL) at a dilution of 1:64 for 10% of samples, while significant bactericidal effects were found in 40% of samples at a dilution of 1:16. The MBCs of other bacterial species tested were at dilutions of 1:1 to 1:8 for both PHMB gauze and PHMB foam. PHMB gauze and foam had MBCs ranging from 1:1 to 1:64 while the other antimicrobials had MBCs of 1:1 to 1:4 across bacterial species tested.

## 4. Discussion

The results of the present study suggested that PHMB gauze and foam have a superior bactericidal effect when compared to hypertonic saline, SSC, Manuka honey, local honey, and commercial, food-grade honey. Our null hypothesis was proven in that PHMB was significantly better at bacterial reduction than medical-grade honey and hypertonic saline. Our hypothesis comparing medical-grade honey to commercial honey and local raw honey was mixed as medical-grade honey was better at reducing bacterial numbers than commercial-grade honey, but not better than local raw honey.

The polyhexanide dressings are commonly used to stop bacterial penetration from the environment to the wound [7,9]. A recent study has shown that the polyhexanide-impregnated dressing must be in contact with the bacteria to have a beneficial effect of bacterial reduction [33]. It was determined in this study that the polyhexanide that is impregnated in the various dressings does not elute into the wound exudate to kill bacteria. A significant benefit of the polyhexanides is the effectiveness against both Gram-positive and Gram-negative bacteria as well as multi-drug-resistant bacteria [12,13]. As a broad-spectrum, yet safe, antimicrobial, polyhexanide dressings should be strongly considered when faced with an infected wound.

Silver sulfadiazine is a commonly used topical wound antibiotic. Numerous studies have shown the benefit of the use of SSC [20,21]. It was not as effective as the polyhexanides or raw honey for bacterial reduction in this study. However, the turbidity in the samples made determining the minimal inhibitory concentration difficult.

Hypertonic saline has been suggested for the removal of bacteria and necrotic tissue [14,15,16]. It works exclusively by osmotic action and is relatively non-selective in wound debridement [14]. Hypertonic saline did not perform as well as the polyhexanide dressings in this study. It is possible that the main benefit of hypertonic saline is actually the debridement of necrotic tissue, thereby reducing the number of bacteria present in a wound through mechanical removal rather than the actual bactericidal effect. This supports caution when using hypertonic saline in wounds with healthy granulation tissue [15].

Honey has long been considered beneficial in wound healing [5,13]. Studies have shown the in vitro bactericidal activity against many bacteria, including multi-drug-resistant strains [22,23]. Manuka honey has generally been considered the gold standard for bacterial reduction in wound healing [24,26]. However, recent studies have shown that other sources of honey can also be effective in reducing bacterial numbers in vitro [34]. In this study, raw honey was nearly as effective as Manuka honey in reducing bacterial numbers in vitro. In our study, local honey appeared to have improved bactericidal effects against all bacterial species compared to commercial, food-grade honey and Manuka honey. Perhaps there are antimicrobial properties in local raw honey that are destroyed in the filtering and pasteurization process that commercial, food-grade honey and medical-grade honey go through. The concern with using local raw honey on a wound involves not knowing all of the components that are present. Local honey could be contaminated with pesticides, antibiotics, and bacterial or fungal spores [35]. These potential contaminants could impact the wound-healing process and have negative side effects.

This study had some inherent limitations, one of which is the low number of isolates used in the study. Another limitation is that a wound bed is a complex environment that is not easily replicated in vitro. The wound microenvironment contains peripheral blood (red blood cells, lymphocytes, macrophages, and neutrophils), growth factors, fibroblasts, necrotic tissue, contamination with debris and multiple types of bacteria, fungus, and yeast [36]. The many components that are present in a wound cannot be easily replicated in a laboratory setting. If the natural wound environment were replicated, this would result in more variables when completing a study of bacterial reduction and would naturally confound results. While it is understood that infection delays wound healing, bacterial reduction cannot be directly correlated with faster wound healing due to the dynamic nature of the wound microenvironment. One major benefit to honey is that it maintains a moist wound environment [22,25]. Moisture is known to help prevent tissue dehydration and cell death, accelerate angiogenesis, increase breakdown of necrotic tissue and fibrin, and potentiate the interaction of growth factors with target cells [37]. These positive properties cannot be observed in vitro. Honey also provides a protective barrier with high viscosity that could help prevent new infections and associated inflammation [22]. These properties of moisture and viscosity contribute to the positive wound-healing effects of honey, but these properties could not be fully considered in this study. Honey seems to have fewer cytotoxic effects and has bactericidal properties that would improve wound healing [14,25]. Honey also maintains a moist wound environment, which we know is vital to proper healing [22].

Interestingly, the majority of antimicrobials exhibited stronger inhibitory and bactericidal activity against a *Streptococcus zooepidemicus* isolate obtained from a wound compared to other bacteria that were ATCC lab-acclimated. There could be differences between wild-type and lab-acclimated bacteria that result in greater antimicrobial susceptibility in wild-type bacteria, but the mechanism is unknown. A comparative study of bacterial reduction using ATCC and wild-type bacteria from the same species could help answer this question.

It is the authors’ hope that the data gathered can help spark additional research regarding the efficacy of wound dressings for different bacterial isolates. Additional research in the chemical composition and contaminants in local raw honey compared to medical-grade Manuka honey would be interesting. Additional research comparing wild-type and ATCC lab-acclimated bacterial isolates is needed to discern whether differences in antimicrobial susceptibility exist. Additional studies looking at a larger number of bacterial isolates would be interesting and useful for clinicians. Based on our data, honey is similar in efficacy to hypertonic saline, so depending on the wound environment and the need for debridement, practitioners may be able to replace hypertonic saline with medical-grade Manuka honey in some circumstances. Additional in vivo studies would help determine the wound-healing capabilities of honey compared to commonly used wound dressings like PHMB gauze and foam, SSC, and hypertonic saline. While an in vivo prospective controlled study would provide less variability, this raises some animal welfare concerns.

## 5. Conclusions

Proper topical wound management is essential to reduce the side effects of systemic antimicrobials, accelerate healing, and reduce antimicrobial resistance [14]. PHMB dressings provide the most effective way to reduce bacterial numbers. They are commercially available and should be strongly considered for use in wound infections. Hypertonic saline seems best suited for wound cases with extensive necrotic tissue. Local honey should be used with caution in vivo due to the risk of contamination.

## Figures and Tables

**Table 1 animals-14-00776-t001:** Minimum Inhibitory Concentration.

Bacteria/ Concentration	Manuka Honey	Store Honey	Local Honey	PHMB Gauze	PHMB Foam
*Pseudomonas* *aeruginosa* 10^5^	1:1	1:1	1:1	1:2	1:1
*Pseudomonas* *aeruginosa* 10^8^	1:1	1:1	1:1	1:4 (90%) 1:2 (10%)	1:1
*Staphylococcus* *aureus* 10^5^	1:1	1:1	1:1	1:16	1:8 (80%) 1:4 (20%)
*Staphylococcus* *aureus* 10^8^	1:1	1:1	1:1	1:8	1:8 (45%) 1:4 (55%)
*Escherichia coli* 10^5^	1:1	1:1	1:1	1:32 (60%) 1:16 (40%)	1:16 (40%) 1:8 (60%)
*Escherichia coli* 10^8^	1:1	1:1	1:1	1:2	1:4 (70%) 1:2 (30%)
*Streptococcus equi* ssp. *Zooepidemicus* 10^5^	1:2 (80%) 1:1 (20%)	1:2 (30%) 1:1 (70%)	1:2 (80%) 1:1 (20%)	1:64 (30%) 1:32 (50%) 1:16 (20%)	1:32 (70%) 1:16 (30%)
*Streptococcus equi* ssp. *Zooepidemicus* 10^8^	1:1	1:1	1:1	1:16 (10%) 1:8 (30%) 1:4 (60%)	1:16 (10%) 1:8 (30%) 1:4 (60%)

1:1 = original concentration of the antimicrobial, 1:2 = 50% original concentration, 1:4 = 25% original concentration, etc. (%) = percentage of samples where bacteria were inhibited at the dilution listed.

**Table 2 animals-14-00776-t002:** Minimum Inhibitory Concentration.

Bacteria/ Concentration	Sterile Gauze	20% Sugar	20% Saline	1% SSC
*Pseudomonas* *aeruginosa* 10^5^	1:1	1:1	1:1	NE
*Pseudomonas* *aeruginosa* 10^8^	1:1	1:1	1:1	NE
*Staphylococcus* *aureus* 10^5^	1:1	1:1	1:1	NE
*Staphylococcus* *aureus* 10^8^	1:1	1:1	1:1	NE
*Escherichia coli* 10^5^	1:1	1:1	1:2 (80%) 1:1 (20%)	NE
*Escherichia coli* 10^8^	1:1	1:1	1:1	NE
*Streptococcus equi* ssp. *Zooepidemicus* 10^5^	1:2 (10%) 1:1 (90%)	1:1	1:1	NE
*Streptococcus equi* ssp. *Zooepidemicus* 10^8^	1:1	1:1	1:1	NE

1:1 = original concentration of the antimicrobial, 1:2 = 50% original concentration, 1:4 = 25% original concentration, etc. (%) = percentage of samples where bacteria were inhibited at the dilution listed.

**Table 3 animals-14-00776-t003:** Minimum Bactericidal Concentration.

Bacteria/ Concentration	Manuka Honey	Store Honey	Local Honey	PHMB Gauze	PHMB Foam
*Pseudomonas* *aeruginosa* 10^5^	No MBC	No MBC	1:1	1:4 (78%) 1:2 (11%) 1:1 (11%)	1:1
*Pseudomonas* *aeruginosa* 10^8^	No MBC	No MBC	1:1	1:2 (78%) 1:1 (22%)	1:1
*Staphylococcus* *aureus* 10^5^	No MBC	No MBC	1:1	1:8 (50%) 1:4 (40%) 1:2 (10%)	1:4 (80%) 1:2 (20%)
*Staphylococcus* *aureus* 10^8^	No MBC	No MBC	No MBC	1:4 (29%) 1:2 (57%) 1:1 (14%)	1:2 (89%) 1:1 (11%)
*Escherichia coli* 10^5^	No MBC	No MBC	1:1	1:32 (60%) 1:16 (30%) 1:8 (10%)	1:8 (60%) 1:4 (40%)
*Escherichia coli* 10^8^	No MBC	No MBC	No MBC	1:2 (90%) 1:1 (10%)	1:4 (70%) 1:2 (40%) 1:1 (20%)
*Streptococcus equi* ssp. *Zooepidemicus* 10^5^	1:1	1:2 (10%) 1:1 (90%)	1:2 (10%) 1:1 (90%)	1:64 (10%) 1:32 (30%) 1:16 (40%) 1:8 (20%)	1:32 (10%) 1:16 (20%) 1:8 (50%) 1:4 (20%)
*Streptococcus equi* ssp. *Zooepidemicus* 10^8^	1:1	1:1	1:1	1:16 (10%) 1:8 (30%) 1:4 (60%)	1:16 (10%) 1:8 (30%) 1:4 (50%) 1:1 (10%)

1:1 = original concentration of the antimicrobial, 1:2 = 50% original concentration, 1:4 = 25% original concentration, etc. (%) = percentage of samples where bacteria were killed at the dilution listed.

**Table 4 animals-14-00776-t004:** Minimum Bactericidal Concentration.

Bacteria/ Concentration	Sterile Gauze	20% Sugar	20% Saline	1% SSC
*Pseudomonas* *aeruginosa* 10^5^	No MBC	No MBC	1:1	NE
*Pseudomonas* *aeruginosa* 10^8^	No MBC	No MBC	No MBC	1:2
*Staphylococcus* *aureus* 10^5^	1:1	No MBC	1:1	NE
*Staphylococcus aureus* 10^8^	1:1	No MBC	No MBC	1:4
*Escherichia coli* 10^5^	1:1	No MBC	1:1	NE
*Escherichia coli* 10^8^	No MBC	No MBC	No MBC	1:1
*Streptococcus equi* ssp. *Zooepidemicus* 10^5^	1:1	1:1	1:2 (20%) 1:1 (80%)	NE
*Streptococcus equi* ssp. *Zooepidemicus* 10^8^	1:1	1:1	1:2 (20%) 1:1 (80%)	1:1

1:1 = original concentration of the antimicrobial, 1:2 = 50% original concentration, 1:4 = 25% original concentration, etc. (%) = percentage of samples where bacteria were killed at the dilution listed.

## Data Availability

Data are contained within the article.

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
