# Peer review of "Comparison of In Vitro Bacterial Susceptibility to Common and Novel Equine Wound Care Dressings"

_animals, 2024, doi:10.3390/ani14050776_

Round 1

Reviewer 1 Report

Comments and Suggestions for Authors

The manuscript has an interesting theme and methodology. However, there is a serious shortcoming in the lack of bibliographical references (only 15). There are numerous recent articles on multiresistance in horses (particularly in wounds, the oral cavity, etc.), the evaluation of topical agents (such as chlorhexidine gluconate and povidone iodine) in equine wounds, the importance of multiresistance and the comparison of various agents in in vivo and in vitro studies, which should be mentioned and discussed. With the structure of the discussion, this manuscript is more like a short communication.

Abstract

Please make some reference to background knowledge present on the literature and just after that present the aim of the work and the methods/results.

Introduction

Lines 52-59: No references have been added to this paragraph. I would advise authors to start the introduction with a review of existing knowledge on the subject, and only at the end of the introduction to include the objectives of the work. Thus, the text on lines 55-59 should come before the text on lines 102-104

Line 62: “USDA APHIS” put the full name of this acronym.

Line 64: “and decrease morbidity”. Please add references.

Line 67: “…post- surgical dressing”. The reference at the end of the phrase in not in square brackets as requested on the instructions for authors.

I think the introduction could be better organized. For example, in the first paragraph, more general topics could be covered, such as the importance of wounds in horses, the importance of topical medicines in equine clinics (wounds), the importance of topical medicines in combating bacterial multiresistance, the importance of bacterial multiresistance particularly in the horse (what are the implications of that), mentioning articles that have already studied that. At this point the authors can speak generically without mentioning the drugs tested in this work. Then, there should be a separate paragraph for each of the compounds tested (PHMB, honey etc). Furthermore, in the paragraph on honey, not only manuka honey should be mentioned, but also homemade and commercial honey.

Line 93: Please mention the compounds present on the honey that have antimicrobial effect.

Results

Tables 1 and 2 are quite difficult to read due to the size of the letters. In addition, they should preferably be structured with Microsoft Word tools so that they are searchable in the text.

Discussion

Once again, there is a great lack of bibliographical references and a tendency to repeat and mention the results obtained without really discussing them with the existing literature.

Lines 195-209: An entire paragraph that just repeats results, gives no references and presents supposed justifications for some results without substantiating them. Please rephrase these gaps throughout the discussion

Lines 218-228: This whole paragraph would have been better in the introduction where the antibacterial characteristics of honey are poorly explained. Please put it in the introduction and not in the discussion

Lines 229-240: Once again, this paragraph doesn't discuss the results, it just gives an introduction to what is known about manuka honey. This kind of text is part of the introduction, not the discussion

Conclusions:

The conclusions are too long. This section is intended only to summarize as much as possible the relevant findings of the study and their importance for researchers and clinicians. I advise authors not to mention methods, objectives, prior knowledge, etc. Only the clinical and research relevance of the study and, perhaps, future studies that they consider interesting.

Author Response

The manuscript has an interesting theme and methodology. However, there is a serious shortcoming in the lack of bibliographical references (only 15). There are numerous recent articles on multiresistance in horses (particularly in wounds, the oral cavity, etc.), the evaluation of topical agents (such as chlorhexidine gluconate and povidone iodine) in equine wounds, the importance of multiresistance and the comparison of various agents in in vivo and in vitro studies, which should be mentioned and discussed. With the structure of the discussion, this manuscript is more like a short communication. References have been increased to 37 and the discussion has been expanded.

Abstract

Please make some reference to background knowledge present on the literature and just after that present the aim of the work and the methods/results. I have added this sentence “Antimicrobial resistance is becoming a problem of concern in the veterinary field, necessitating the use of effective topical treatments to aid healing of wounds. Honey has been used for thousands of years for its medicinal properties, but in recent years medical grade Manuka honey is being used to treat infected wounds.”

Introduction

Lines 52-59: No references have been added to this paragraph. I would advise authors to start the introduction with a review of existing knowledge on the subject, and only at the end of the introduction to include the objectives of the work. Thus, the text on lines 55-59 should come before the text on lines 102-104 I have made these changes, added references, and re-ordered the text. The entire paragraph is changed to “There is a high incidence of traumatic wound infections in equine patients and an increasing prevalence of antimicrobial resistant bacterial species in wounds [1,2]. A 2015 report published by the United States Department of Agriculture Animal and Plant Health Inspection Service (USDA APHIS) indicated that 16.3% to 27.8% of equid deaths (20 years of age and less) nation-wide were the result of skin wounds or trauma [3]. A 2018 prospective study on antimicrobial resistance in horses in the UK showed that prevalence of multi-drug resistance for E.coli was 31.7% and for Staphylococcus spp it was 25.3% [4]. The nature of horses and their habitat puts them at risk for wounds that are likely to become contaminated with soil bacteria [5]. Horses often get wounds on their legs and these wounds are more predisposed to delayed healing due to tension and infection [5, 6]. Overuse of antibiotics is a large factor contributing to increasing bacterial resistance [5]. Researchers are investigating alternative therapies to help reduce the need for systemic antibiotics in both human and animal wound infections [5]. Topical treatments reduce the development of resistance to systemic antibiotics, are cost-effective, and decrease morbidity [7, 8]. Topical antimicrobials affect multiple targets and reduce the use of prophylactic antibiotics, and avoid side effects commonly seen with systemic [7, 8]. A better understanding of topical antimicrobial agents in animals is important to provide the most successful wound healing.”

Line 62: “USDA APHIS” put the full name of this acronym. “A 2015 report published by the United States Department of Agriculture Animal and Plant Health Inspection Service (USDA APHIS) indicated that 16.3% to 27.8% of equid deaths (20 years of age and less) nation-wide were the result of skin wounds or trauma [3].”

Line 64: “and decrease morbidity”. Please add references. “Topical treatments reduce the development of resistance to systemic antibiotics, are cost-effective, and decrease morbidity [7, 8].”

Line 67: “…post- surgical dressing”. The reference at the end of the phrase in not in square brackets as requested on the instructions for authors. This formatting has been edited.

I think the introduction could be better organized. For example, in the first paragraph, more general topics could be covered, such as the importance of wounds in horses, the importance of topical medicines in equine clinics (wounds), the importance of topical medicines in combating bacterial multiresistance, the importance of bacterial multiresistance particularly in the horse (what are the implications of that), mentioning articles that have already studied that. At this point the authors can speak generically without mentioning the drugs tested in this work. Then, there should be a separate paragraph for each of the compounds tested (PHMB, honey etc). Furthermore, in the paragraph on honey, not only manuka honey should be mentioned, but also homemade and commercial honey. I have re-structured and expanded the entire introduction based on this reviewers suggestions with the exception of much discussion on homemade and commercial honey as it was challenging to find references to studies on this topic. We studied them out of interest for comparison.

Line 93: Please mention the compounds present on the honey that have antimicrobial effect. Line 122 . “The efficacy of honey for wounds can be attributed to its bactericidal compounds, moist environment that promotes healing, and high viscosity that maintains an effective barrier [22]. Antimicrobial properties of honey in general include hypertonicity, low pH, and hydrogen peroxide production, but these properties are negligible with dilution [22].  Phenolics are compounds found in honey that act as antioxidants and contribute to anti-inflammatory and healing properties of honey [24].”

Results

Tables 1 and 2 are quite difficult to read due to the size of the letters. In addition, they should preferably be structured with Microsoft Word tools so that they are searchable in the text. I have revised all the tables so they are in microsoft word and more easily readable. Please see updated manuscript.

Discussion

Once again, there is a great lack of bibliographical references and a tendency to repeat and mention the results obtained without really discussing them with the existing literature. I have increased the references and expanded on the discussion. Please see updated manuscript.

Lines 195-209: An entire paragraph that just repeats results, gives no references and presents supposed justifications for some results without substantiating them. Please rephrase these gaps throughout the discussion I have left this paragraph largely as is since it feels like a relevant introduction to the discussion section but I have expanded on the discussion. Please see updated manuscript.

Lines 218-228: This whole paragraph would have been better in the introduction where the antibacterial characteristics of honey are poorly explained. Please put it in the introduction and not in the discussion I have moved this paragraph into the introduction.

Lines 229-240: Once again, this paragraph doesn't discuss the results, it just gives an introduction to what is known about manuka honey. This kind of text is part of the introduction, not the discussion. I have moved this paragraph into the introduction. Please see updated manuscript.

Conclusions:

The conclusions are too long. This section is intended only to summarize as much as possible the relevant findings of the study and their importance for researchers and clinicians. I advise authors not to mention methods, objectives, prior knowledge, etc. Only the clinical and research relevance of the study and, perhaps, future studies that they consider interesting. Conclusions have been reduced and updated as follows “Proper topical wound management is essential to reduce side effects of systemic antimicrobials, speed healing, and reduce antimicrobial resistance [14]. PHMB dressings provided the most effective way to reduce bacterial numbers. They are commercially available and should be strongly considered for use in wound infections. Hypertonic saline seems best suited for wound cases with extensive necrotic tissue. Local honey should be used with caution in-vivo due to the risk for contamination.”

Reviewer 2 Report

Comments and Suggestions for Authors

The manuscript "Comparison of in-vitro bacterial susceptibility to common and novel equine wound care dressings" focus an important and common subject related to equine veterinary practice, and could help practitioners how to best manage an infected wound.

The Simple Summary and Abstract are correct and elucidates the content of the manuscript.

Lines 24-25 – “… four common equine wound pathogens”. I suggest to elucidate the species.

four common equine wound pathogens (Streptococcus zooepidemicus; etc…)

Line 28 “to other commonly used wound dressings.” I also recommend to specify which wound dressings.

The introduction section is satisfactory.

Line 67 – “…post-surgical dressing.1”. Please remove 1 at the end of the sentence or use appropriate reference.

Line 77 – “normal skin flora.a”. Please remove a at the end of the sentence or use appropriate reference.

The materials and methods seem adequate.

 Line 115 “…silver sulfadiazine cream,” – specify concentration. 1%?

The results are sound, even thought the low number of isolates tested.

The discussion is correct, but the authors should emphasize the low number of isolates tested…

It would be also interesting to know if the isolates used were sensible or resistant to frequent antimicrobials used. Can the authors provide as supplementary information, the antimicrobial susceptibility tests of the bacteria used?

Author Response

The manuscript "Comparison of in-vitro bacterial susceptibility to common and novel equine wound care dressings" focus an important and common subject related to equine veterinary practice, and could help practitioners how to best manage an infected wound.

The Simple Summary and Abstract are correct and elucidates the content of the manuscript.

Lines 24-25 – “… four common equine wound pathogens”. I suggest to elucidate the species.

… four common equine wound pathogens (Streptococcus zooepidemicus; etc…) Updated to “ The pathogens studied include ATCC lab acclimated Pseudomonas aeruginosa, Escherichia coli, and methicillin-resistant Staphylococcus aureus and from an equine sample submitted to the Colorado State Veterinary Diagnostic Laboratory (Streptococcus equi ssp. zooepidemicus (Streptococcus zooepidemicus)).”

Line 28 “to other commonly used wound dressings.” I also recommend to specify which wound dressings. Updated to “to other commonly used wound dressings (20% hypertonic saline, silver sulfadiazine cream, PHMB gauze, PHMB foam).”

The introduction section is satisfactory.

Line 67 – “…post-surgical dressing.1”. Please remove 1 at the end of the sentence or use appropriate reference. I have updated this

Line 77 – “normal skin flora.a”. Please remove a at the end of the sentence or use appropriate reference. I have fixed this error

The materials and methods seem adequate.

 Line 115 “…silver sulfadiazine cream,” – specify concentration. 1%? I have added this 1% throughout the text to clarify the concentration

The results are sound, even thought the low number of isolates tested.

The discussion is correct, but the authors should emphasize the low number of isolates tested… Added” This study had some inherent limitations, one of which is the low number of isolates used in the study” into the discussion

It would be also interesting to know if the isolates used were sensible or resistant to frequent antimicrobials used. Can the authors provide as supplementary information, the antimicrobial susceptibility tests of the bacteria used? We were not able to find this reference, I believe there is not a lot of data on the sensitivity to topical wound management products (except mupirocin for MRSP) when compared to systemic (enteral or parenteral) antibiotics.

Round 2

Reviewer 1 Report

Comments and Suggestions for Authors

I would like to thank the authors for their efforts in incorporating all the corrections. The manuscript is better structured and much easier to read. 

Just a few minor corrections:

Line 97: in order to standardize the text, put the full name of "PHMB" here instead of on line 108.

Line 111: "Hypertonic saline (20%) dressings..." replace with "these dressings..." to avoid repetition with previous line.

Lines 185-191: This part is still a bit confusing. It seems that the hypothesis that the authors want to test and the objective of the work do not coincide. The authors mention that their hypothesis is that Manuka honey has a similar antimicrobial efficacy to PHMB. However, they point out that the aim of the study was to compare the efficacy of different types of honey. I'd like to take this opportunity to mention that PHMB turns out to be more effective and, in my opinion, this fact is given little weight in the discussion. I don't think a substantial improvement/change is needed in the discussion since, although it focuses more on honey than on PHMB, it tries to explain why honey was less effective in vitro, which is fine with me. However, the text of the mentioned lines (185-191) should be reworded.

Lines 314-318: In my opinion this is part of the materials and methods 

Lines 395-396: This is repeated in the text. Delete

Lines 308-310 and 402-404: there is a duplication of sentences. The content is exactly the same. Consider deleting lines 308-310 and leaving 402-404. 

Author Response

I would like to thank the authors for their efforts in incorporating all the corrections. The manuscript is better structured and much easier to read. 

Thank you for your time spent reviewing this paper. My edits are below in red text

Just a few minor corrections:

Line 97: in order to standardize the text, put the full name of "PHMB" here instead of on line 108.

Changed line 97 to “There are many different types of wound dressings impregnated with PHMB (polyhexamethylene biguanide), and these dressings help by decreasing bacterial load and preventing bacterial penetration, therefore reducing colonisation of the wound bed [7].” And removed “polyhexamethylene  biguanide” from line 108

Line 111: "Hypertonic saline (20%) dressings..." replace with "these dressings..." to avoid repetition with previous line.

Updated: “Hypertonic saline (20%) dressings are used to debride infected or exudative wounds [14]. These dressings work by removing fluid from bacteria and necrotic cells using osmotic gradients, allowing bacteria and debris to be lifted off a wound with dressing change [14]. “

Lines 185-191: This part is still a bit confusing. It seems that the hypothesis that the authors want to test and the objective of the work do not coincide. The authors mention that their hypothesis is that Manuka honey has a similar antimicrobial efficacy to PHMB. However, they point out that the aim of the study was to compare the efficacy of different types of honey. I'd like to take this opportunity to mention that PHMB turns out to be more effective and, in my opinion, this fact is given little weight in the discussion. I don't think a substantial improvement/change is needed in the discussion since, although it focuses more on honey than on PHMB, it tries to explain why honey was less effective in vitro, which is fine with me. However, the text of the mentioned lines (185-191) should be reworded.

“Our hypothesis is that medical grade Manuka honey would have comparable bactericidal effect to PHMB and would have better bactericidal effect when compared to commercial, food-grade honey and raw, local honey. The goal of this research was to provide veterinary practitioners with comparative data on the use of medical-grade, commercial, food-grade, and raw local honey as antimicrobial dressings for inhibiting growth of common bacterial species in wounds. We ascertained bacterial susceptibility in vitro to gather information for possible future in vivo utilization.” Reworded to  “The goal of this study was to provide veterinary practitioners with comparative data on the use of various dressings including: PHMB, Hypertonic Saline, Medical-grade honey, commercial food-grade honey, and local raw honey to inhibit growth of common bacterial species in wounds. We ascertained bacterial susceptibility in vitro to gather information for possible future in vivo utilization. Our hypothesis is that medical-grade honey would have comparable bactericidal effect to PHMB and would have better bactericidal effect than commercial food-grade honey and raw local honey. We also hypothesized that hypertonic saline would have similar bactericidal effects to PHMB.”

Added to the discussion line 284-288: “Our hypothesis was proven null in that PHMB was significantly better at bacterial reduction than medical grade honey and hypertonic saline.  Our hypothesis comparing medical-grade honey to commercial honey and local raw hone was mixed as medical-grade honey was better at reducing bacterial numbers than commercial grade honey, but not better than local raw honey.”

Lines 314-318: In my opinion this is part of the materials and methods 

These lines were moved into first paragraph of results, lines 218-222

Lines 395-396: This is repeated in the text. Delete Deleted this line.

Lines 308-310 and 402-404: there is a duplication of sentences. The content is exactly the same. Consider deleting lines 308-310 and leaving 402-404.  Deleted “Most antimicrobials exhibited stronger inhibitory and bactericidal activity with Streptococcus zooepidemicus that was obtained from an equine wound compared to the other organisms that were ATCC lab acclimated.” And left lines 402-404 as is